# The Role of Vitamins in Oral Health. A Systematic Review and Meta-Analysis

**DOI:** 10.3390/ijerph17030938

**Published:** 2020-02-03

**Authors:** Maria Grazia Cagetti, Thomas Gerhard Wolf, Christian Tennert, Nicole Camoni, Peter Lingström, Guglielmo Campus

**Affiliations:** 1Department of Biomedical, Surgical and Dental Science, University of Milan, IT-20142 Milan, Italy; maria.cagetti@unimi.it (M.G.C.); n.camoni@gmail.com (N.C.); 2Department of Restorative, Preventive and Paediatric Dentistry, University of Bern, CH-3010 Bern, Switzerland; christian.tennert@zmk.unibe.ch (C.T.); Guglielmo.campus@zmk.unibe.ch (G.C.); 3Department of Cariology, Institute of Odontology, Sahlgrenska Academy, University of Gothenburg, SE-41390 Gothenburg, Sweden; peter.lingstrom@odontologi.gu.se; 4Department of Surgery, Microsurgery and Medicine Sciences, School of Dentistry, University of Sassari, IT-07100 Sassari, Italy

**Keywords:** dental caries, dental erosion, gingivitis, periodontal disease, vitamin/vitamins

## Abstract

The association between vitamins and oral health have recently been discussed, yielding increased attention from medical and dental perspectives. The present review aimed to systematically evaluate and appraise the most recently scientific papers investigating the role of vitamins in the prevention and treatment of the main oral diseases as hard dental pathological processes and gum/periodontal disease. Randomized controlled trials, cross-sectional studies, cohort studies, comparative studies, validation studies and evaluation studies, following the Preferred Reporting Items for Systematic Reviews and Meta-Analyses (PRISMA) guidelines, reporting associations between vitamins and oral diseases or the use of vitamins to prevent or treat oral diseases in patients of any age were included. PubMed, Embase and Scopus were searched to November 2019 using an ad hoc prepared search string. All the papers meeting the inclusion criteria were subjected to a quality assessment. The search identified 1597 papers; 741 were selected after removing duplicates. A total of 334 articles were excluded after title and abstract evaluation; 407 were assessed and 73 papers were full-text assessed; other 14 papers were discharged after full text evaluation, leaving finally 58 papers included. In general, there is weak evidence supporting the association between vitamins and both gingival/periodontal disease and hard dental pathological processes.

## 1. Introduction

The role of vitamins is well known in a medical perspective, but the scientific evidence regarding the oral health perspective is still not fully clarified [1].

Vitamins are catalysts for all metabolic reactions, using proteins, fats and carbohydrates for energy, growth and cell maintenance. As only small amounts of these fundamental substances are obtained from food, vitamins are often administered though food supplements [2]. Fat-soluble vitamins such as A, C, D, E and K can be stored in the liver and fat tissues as reserves, while water-soluble vitamins as B and C are expelled if not absorbed.

It is general knowledge that vitamins play a significant effect on oral and general health where its imbalance leads to malnutrition. The process of chewing allows one to extract the greatest possible amount of nutrients and the number and distribution of teeth influence the chewing efficacy. The available literature on the role of vitamins toward oral health is really scarce with no available data on the prevalence of oral disease related to vitamin deficiencies. Teeth loss affects dietary choice and nutritional status [3]. A significant improvement of vitamin D levels was obtained in partially dentate patients aged ≥ 65 years after the replacement of lost teeth using prosthetic solutions [4]; still, no strong evidence on the effect of tooth loss on nutritional status was found in a recent review [5].

Vitamin deficiency prompted several non-specific oral conditions as glossitis, stomatitis and mucosal ulceration. Glossitis with linear lesions was postulated to be an early sign of vitamin B12 paucity [6].

Vitamin D deficiency leads to reduced bone density, osteoporosis, and, as consequence, to the progression of periodontal disease; on the other hand, sufficient levels of this vitamin might reduce the risk of gingivitis and periodontitis; the vitamin acting as immunomodulator, anti-inflammatory and antiproliferative agent [7].

In the developmental phases, hard dental tissues are strongly influenced by nutritional status and consequently to vitamin deficiency [8]. A positive relationship between malnutrition, enamel hypoplasia and caries in the primary dentition was postulated in children [9,10]. 

The frequent and prolonged exposure to acidic agents contained in food, beverages, drugs or food supplements can lead to significant tooth wear [11]. Chewable vitamin C tablets have been reported to have a pH of about 2, lower than the critical pH value (5.5) for enamel dissolution, postulating an association between vitamin C and erosion with an odds ratio of 1.16 [12].

The aim of the present study was to perform a systematic review and meta-analysis of the scientific papers published during the last 20 years, investigating the association between vitamins and gingival/periodontal disease and hard dental pathological processes such as dental caries, tooth wear and developmental defects. 

## 2. Materials and Methods

Reporting of this review follows the Preferred Reporting Items for Systematic Reviews and Meta-Analyses (PRISMA) guideline [13]. The review protocol was registered with the International prospective register of systematic reviews (PROSPERO) system (ID 150613, 12 September 2019).

### 2.1. Eligibility Criteria

The review included randomized controlled trials (RCTs), cross-sectional studies, comparative studies, validation studies and evaluation studies, reporting vitamins supplementary (foods, tablets etc) or vitamin serum levels in patients of any age. Only papers in English published from 1 January 2000 to 30 November 2019 were collected. Electronically published articles and paper-based article were taken into consideration.

### 2.2. Information Sources

Electronic databases Medline via PubMed, Embase via Ovid and Scopus were screened for articles.

### 2.3. Information Sources and Search Strategy

Several search strategies were used. The first included a combination of Medical Subject Headings (MeSH) terms and key words: *Vitamin* OR *Vitamins* OR *oral health*, OR *caries* OR *dental caries* OR *periodontal disease* OR *dental erosion* OR *gingivitis*. The second strategy included the search string “*Vitamins* OR *Vitamin* OR *vitamin A* OR *Vitamin B* OR *Vitamin C* OR *Vitamin D* OR *Vitamins B* OR *Vitamin E* OR *Vitamin K*” and “*Oral health* OR *oral health* OR *caries* OR *dental caries* OR *root caries* OR *tooth diseases* OR *salivation* OR *saliva* OR *periodontal diseases* OR ‘*dental erosion*’ OR *tooth erosion* OR *tooth erosion* OR ‘*cariogenic bacteria’* OR *biofilms* OR *biofilm* OR *periodontitis* OR *periodontitis* OR *gingivitis* OR *gingivitis* OR *dental plaque* OR *plaque*”. Cross-referencing was performed using the bibliographies of full-text articles. Grey literature was also retrieved via opengrey.eu (http://www.opengrey.eu).

### 2.4. Study Selection

Repeated or duplicate papers were excluded after comparing the results from the different research strategies. Three authors (T.G.W., M.G.C., and N.C.) independently examined all the abstracts of the papers. All the papers meeting the inclusion criteria were obtained in the full-text format. The authors independently assessed the papers to establish whether each paper should or should not be included in the systematic review.

### 2.5. Data Collection, Summary Measures and Synthesis of Results

Data collection and synthesis was independently carried out by three authors (G.C., M.G.C. and N.C.) using an ad hoc designed data extraction form, without masking journal title or authors. Different studies outcomes were compared on the use of vitamins to prevent or treat oral diseases per different diseases and publication years. To facilitate the synthesis, the results were summarised in tables where each selected paper was included and the main aspects presented (i.e., vitamin and oral disease studies, sample, age, healthy subjects or affected by systemic diseases effect on the disease, statistically significance). For each paper, the following data were searched and recorded when available: a) publication year and study duration; b) details/characteristics of the participants at baseline; c) oral data, including gingival or periodontal conditions or gingival bleeding or pocket dept or gingival recession or loss of clinical attachment level; actual caries status, caries experience and caries increment measured through DMFT/S or dmft/s (for decayed, missing, filled teeth/surfaces in permanent and primary teeth indexes) or ICDAS (for International Caries Detection and Assessment System), or other detection systems; the presence of tooth wear; the presence of developmental enamel defect.

The ProMeta 3 Software (IdoStatistics https://idostatistics.com/prometa3/, Cesena, Italy: Internovi) was used for the meta-analysis of the data. Mean difference (MD) and odds ratio (OR) were chosen for calculating the effect size. The analysis was computed on the different vitamins used. Associations between vitamins and gingivitis, periodontitis, caries and enamel defects were computed separately. The I² statistic was calculated to describe the percentage of variation across studies due to heterogeneity rather than chance [14]. The heterogeneity was categorized as follows: <30% not significant; 30–50% moderate; 51–75% substantial, and 76–100% considerable. Whether homogeneity was obtained or not, the random effects model (REM) with 95% confidence intervals was chosen as the meta-analysis model. Potential moderators as publication type, publication year, age groups, vitamins were evaluated and analysed to explain which factors might affect heterogeneity. The funnel plot method was used to assess the potential role of publication bias [15]. The significance levels of the effect sizes were determined based on the two-tailed test. In all tests, the level of significance was set at *p* < 0.05. 

### 2.6. Assessment of Bias across Studies

The risk of bias assessment was conducted by two authors (C.T., T.G.W.). The methodological quality of the included RCTs was scored according to the customized quality assessment tool developed by the National Heart, Lung, and Blood Institute and Research Triangle Institute International for Observational Cohort and Cross-Sectional Studies and Study Quality Assessment Tools Guidance for Assessing the Quality of Controlled Intervention Studies www.nhlbi.nih.gov/health-topics/study-quality-assessment-tools [https://www.nhlbi.nih.gov/health-topics/study-quality-assessment-tools]. The tools included items for evaluating potential flaws in study methods or implementation, including sources of bias (e.g., patient selection, performance, attrition, and detection), confounding, study power, the strength of causality in the association between interventions and outcomes and other factors. For each item, “yes,” “no,” or “cannot determine/not reported/not applicable” was selected. Each study was finally scored as “good” when it has the least risk of bias, “fair” if it is susceptible to some bias and "poor" when significant risk of bias is conceivable. 

Disagreements between authors were resolved by discussion. Where this was not possible, another author was consulted (M.G.C.).

## 3. Results

The search identified 1597 papers; 741 were selected after removing duplicates. A total of 334 articles were excluded after title and abstract evaluation; 407 were assessed and 73 papers were full-text assessed (Appendix A. List of excluded papers); the quality assessment scores of the papers included is presented in the Appendix A, other 14 papers were discharged after full text evaluation (Appendix A. List of excluded papers after full text evaluation), leaving 58 included papers (Figure 1).

Forty papers concerned on gingival/periodontal disease and 20 (two papers were in common) on hard dental pathological processes were included. Regarding gingival/periodontal disease, 26 papers were ranked of as being of good quality, 12 were classified of fair quality and only two of poor quality. Regarding hard dental tissues, 16 papers were ranked of as being of good quality, four were classified of fair quality and only two of poor quality (Table 1).

Funnel plot analysis (Figure 2) showed that for gingivitis, caries and enamel defects no study was trimmed, and the overall effect sizes observed and estimated were the same 0.81, (95% CI ranging from 0.33 to 1.29; *p* = 0.06) and 1.04, (95% CI ranging from 0.92 to 1.18; *p* = 0.52) and 0.27 (95% CI ranging from −0.04 to 0.57; *p* = 0.09) respectively. Furthermore, no significant publication bias existed based on the Egger regression analysis (*p* = 0.109, 0.79 and 0.19, respectively). Regarding periodontal disease, six studies were trimmed, the observed effect size was 0.97, (95% CI ranging from 0.78 to 1.219; *p* = 0.78) while the estimated one was 0.76, (95% CI ranging from 0.60 to 0.97; *p* = 0.03) with no statistically significant publication bias (*p* = 0.91).

Due to the low numbers of studies for each vitamin, the heterogeneity was very high for all vitamins ranging from 83.68% for vitamin B to 99.13% for vitamin C (Figure 3). Regarding gingivitis, the heterogeneity analysis was measured as considerable with the highest value observed for vitamin C. Heterogeneity analysis for periodontal disease revealed the highest value for vitamin B (97.39%) followed by vitamin D (84.39%) and then vitamin C (13.27%). Heterogeneity analysis for caries showed the highest value for vitamin C (95.50%) while a substantial I^2^ value was observed for vitamin D (70.06%). Considering enamel defects, there were not enough data levels for performing this analysis. 

### 3.1. Gingival/Periodontal Disease

The main characteristics of the included studies regarding gingivitis and periodontitis/tooth loss are reported in Table 2. 

Four studies were conducted to evaluate the effect of vitamin D on gingivitis; in one study vitamin D was given alone [30], while in the other three it was administered in combination with vitamin C [23] or vitamin B and E [18] or vitamin A, B1, B2, B6, B9, C, E [38] through the diet. A dose-dependent effect was found on gingival scores, showing the supplementation of 2000 International Unit (IU) of vitamin D obtained a greater improvement in gingival parameters compared to lower amount (1000 IU and 500 IU). A similar effect was obtained with a 4-week diet rich in vitamin C, D, Omega-3 fatty acids and antioxidants. All inflammatory parameters (gingival index, bleeding on probing and the total periodontal inflamed surface area) were halved compared to baseline. The administration of a dietary supplement containing different micronutrients (including vitamin D, C, E, B complex) for 3 months produced a slight improvement of the gingival inflammation in students under stress with poor oral hygiene, compared to students also under stress but not provided with the dietary supplement. The 6-month administration of a dietary supplement containing vitamin A, B1, B2, B6, B9, C, D, E in Type 2 diabetic adults, reduced gingivitis and oral ulcers incidence compared to placebo (*p* < 0.05).

Five studies analyzed the effect of vitamin C on gingival parameters, three of them considering vitamin C as the only variable [24,29,49] and two on vitamin C combined with other vitamins [41,55]. All these studies used different administration modalities, including toothpaste, dietary supplement, chewing gum and foods. In the first three studies, vitamin C showed to reduce gingival scores of inflammation. vitamin C and B9 levels were statistically associated to bleeding on probing (*p* < 0.01) [41]; vitamin A was also associated (*p* < 0.05), while vitamin B1 and B2 levels were found to be associated to gingivitis presence in adolescent girls, while vitamin A and B3 resulted in not being associated [55]. Three studies investigated the effects of vitamin B9 on gingival scores, two with the vitamin as the only variable [30,34] and one with vitamin B9 combined with vitamin B12 [48]. Vitamin B9 was administered in patients with epilepsy to reduced Phenytoin-induced gingival overgrowth (PIGO) [23,24,29,30,34,38,39,41,49,55]. In both studies vitamin B9 administration reduced the development of PIGO or delayed its onset. A statistically significantly association between vitamin B9 and gingival index was found in smokers (*p* < 0.01) compared to non-smokers, while vitamin B12 resulted not associated [48]. Finally, a fluoridated toothpaste containing vitamin B3 and pro-vitamin B5 provided a statistically significantly reduction in calculus presence compared to a fluoridated toothpaste not containing vitamins (*p* = 0.01) [43].

Twelve papers were concerned on the effect of vitamin D on periodontitis. A reduction of the clinical disease level (i.e., clinical attachment level and/or probing pocket depth) was described in five papers [17,19,23,45,52], while in four papers [20,25,33,35] vitamin D levels had no statistically significant impact on clinical attachment level and probing pocket depth improvements in teriparatide patients. Low serum vitamin D levels were not statistically associated to periodontitis and tooth loss in pregnant and post-menopausal women [22,32,36].

Four papers concerned on the effect of vitamin C on periodontitis. Two papers [29,51] underlined the reduction of gingival bleeding consequent to use of vitamin C in patients affected by chronic periodontitis. The use of fruit or vegetables rich in vitamin C was statistically significantly lower in subjects affected by chronic periodontitis respect to healthy subjects [52]. Serum concentrations of vitamin C, bilirubin, and total antioxidant capacity were inversely associated with periodontitis, the association being stronger in severe disease [46]. Vitamin B-complex supplement resulted in statistically significantly superior clinical attachment gains and reduction of inflammatory mediators respect to placebo [39,50]. The use of a standard multivitamin formula provided modest benefits in reducing periodontal inflammation [37].

Four studies reported on gingivitis/periodontitis/tooth loss and vitamin D during particular periods of a woman’s life, pregnancy [21,36], menopause [22,32]. Low vitamin D levels in saliva and serum were statistically associated with gingivitis and periodontitis during pregnancy [21,36]. vitamin D in post-menopausal was statistically associated with periodontitis [32], but the association with tooth loss failed [22,32]. 

### 3.2. Hard Dental Pathological Processes 

The main characteristics of the studies included regarding hard tooth tissues (caries and enamel defects) are reported in Table 3.

Eleven papers were focused on vitamin D and caries; six of them [10,22,57,58,68] were observational studies showing a statistically significantly association between vitamin D serum level and caries level and/or experience. Five papers [10,57,58,68] were on children (age range 1–11 years).

Vitamin D treatment in children or in mothers during pregnancy were associated to caries incidence or experience in five papers [60,61,64,65,66]. Two cross-sectional studies [67,71] were done associating vitamin C intake and caries (levels and experience) and erosion in children. Multivitamins intake was related to caries in two papers [67,70]; vitamin B2, B7, B12 were associated to caries [67], while vitamin A was not statistically significantly correlated and vitamin C and vitamin E. statistically significantly correlated to caries [70]. In early childhood (up to 8 years), serum levels of vitamin D seem to be associated with DMFT and caries risk in the following years [10,58,63]. In early teenagers (10–11 years old) a significantly less caries experience of the first molars was found, when serum vitamin D levels are higher than 50 nmol [57]. Another study found a direct correlation of serum vitamin D levels in children, 6–17 years of age. The authors found a drop in DMFT of 0.66 at each 10 ng/ml increase of vitamin D [60]. Regarding the correlation of vitamin C and the occurrence of caries lesions there are controversial results. One study found vitamin C supplementation, but also soft drink consumption, to be positively correlated to caries in 12-year-old children [67]. Another study including 6- to 13-year-old children found negative correlations between vitamins C and E and caries risk. Salivary vitamin A levels are not to be statistically significantly associated to caries risk [56]. High intake of especially vitamin B12, riboflavin, pantothenic acid and nicotinic acid seem to be correlated to lower caries rates in 5 years old children, but the association seems not clinically significant [70].

Despite caries, the occurrence of enamel hypoplasia seems to be associated with low blood levels of vitamin D during pregnancy [62], whereas the occurrence of MIH seems not affected by fetal, postnatal and early childhood levels of vitamin D [59]. One study found no association between the occurrence of enamel defects and vitamin D in 1- and 2-year-old children born preterm [72].

Newly erupted permanent teeth of children have immature enamel, which is more susceptible to acid attack of nutritional acids, e.g. soft drinks or fruit juices. The intake of vitamin C supplements was found to be associated with the incidence of erosive tooth wear in early childhood. In 2- to 5-year-old children, vitamin C supplementation significantly reduced the incidence of erosive tooth wear [71]. A study on 10- to 12-year-old children found that an intake of vitamin supplements (not specified) seems not to affect the incidence of erosive tooth wear, but decreased their progression significantly [69]. In general, malnutrition and associated deficiency in vitamin intake increases the occurrence of enamel hypoplasia in children [70]. 

## 4. Discussion 

There is no clear scientific evidence on the role played by vitamins on oral health. There is a general consensus on the effect of vitamins deficiencies or supplementation on oral health but without a substantial scientific evidence. 

The aim of this systematic and metanalysis review was to evaluate if there were associations between vitamin intake (supplementation or diet intake or saliva/serum level) and gingival/periodontitis and hard dental pathological processes (dental caries, tooth wear and enamel defects). 

The lack of convincing associations and the relative dearth of possible associations suggest that the evidence for oral health benefits of vitamins that may be reaped from population-wide vitamin supplementation is weak. The issues to attain positive outcomes from experimental clinical trials are linked to the dosages of vitamins or more effective treatments that might act as confounding factors, thereby camouflaging the effect of the vitamins. Probable associations, where highly significant effects appear in randomised trials, hold the most promise for clinical translation; however, studies pertain to specific populations (children, pregnant women, patients with systemic diseases), and even in these cases the evidence is not sufficient to make universal recommendations about daily intake. Multivitamin supplement or a combination of two or more vitamins adds more biases as it is not possible to identify the single benefit of each vitamin. 

Moreover, the majority of papers are short-term papers. Hence, it was not possible to provide clear scientific evidence for the role played by vitamins. Concerning observational studies, there was a wide variety in the use of dietary supplement and clinical parameters used, which could explain the differences found among their results. 

Until at least the middle of the 18th century, several oral diseases like periodontitis were considered a manifestation of vitamin deficiency [1,8,73], but there is no sufficient data supporting the need for vitamin supplementation for oral health. Vitamin D has been related to gingival inflammation [47] and tooth loss [17,19,20,25,33,35,36]. Moreover, vitamins and in particular vitamin D as a promising oral health-preventive agent were the object of several previous reviews [73,74,75,76,77], systematic [74,75,76] and narrative [73,77] leading to a low-certainty conclusion that vitamins may reduce the incidence of caries and periodontitis.

## 5. Conclusions

In general, although the existing literature suggests that vitamins are important in the prevention and treatment of oral diseases, there is weak evidence supporting the association between vitamins and both gingival/periodontal disease and hard dental pathological processes. 

Overall, future longitudinal studies of the oral outcomes associated with vitamins and focused research on the detailed biological mechanisms will have broader applications in dentistry and medicine.

## Figures and Tables

**Figure 1 ijerph-17-00938-f001:**
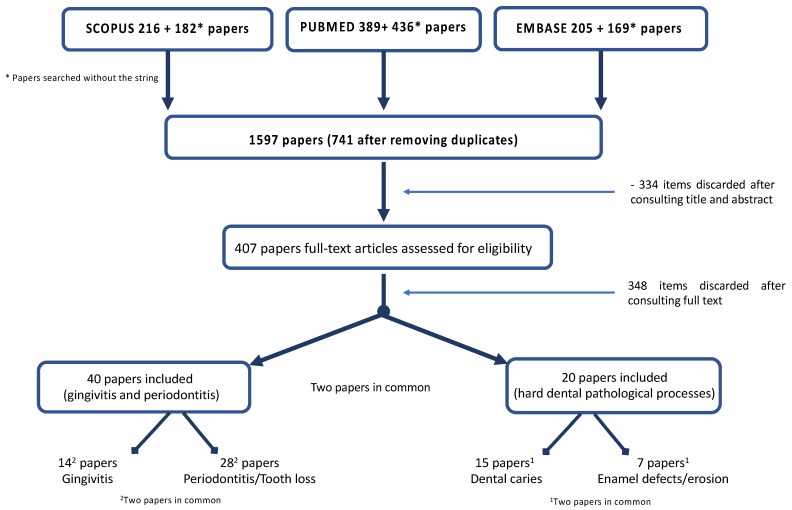
Flow chart of the search.

**Figure 2 ijerph-17-00938-f002:**
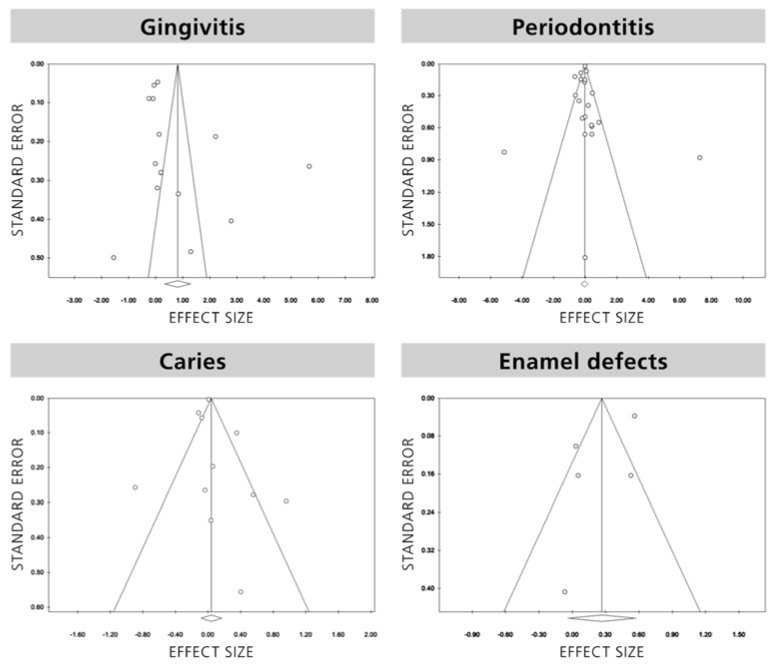
Funnel plots of publication bias.

**Figure 3 ijerph-17-00938-f003:**
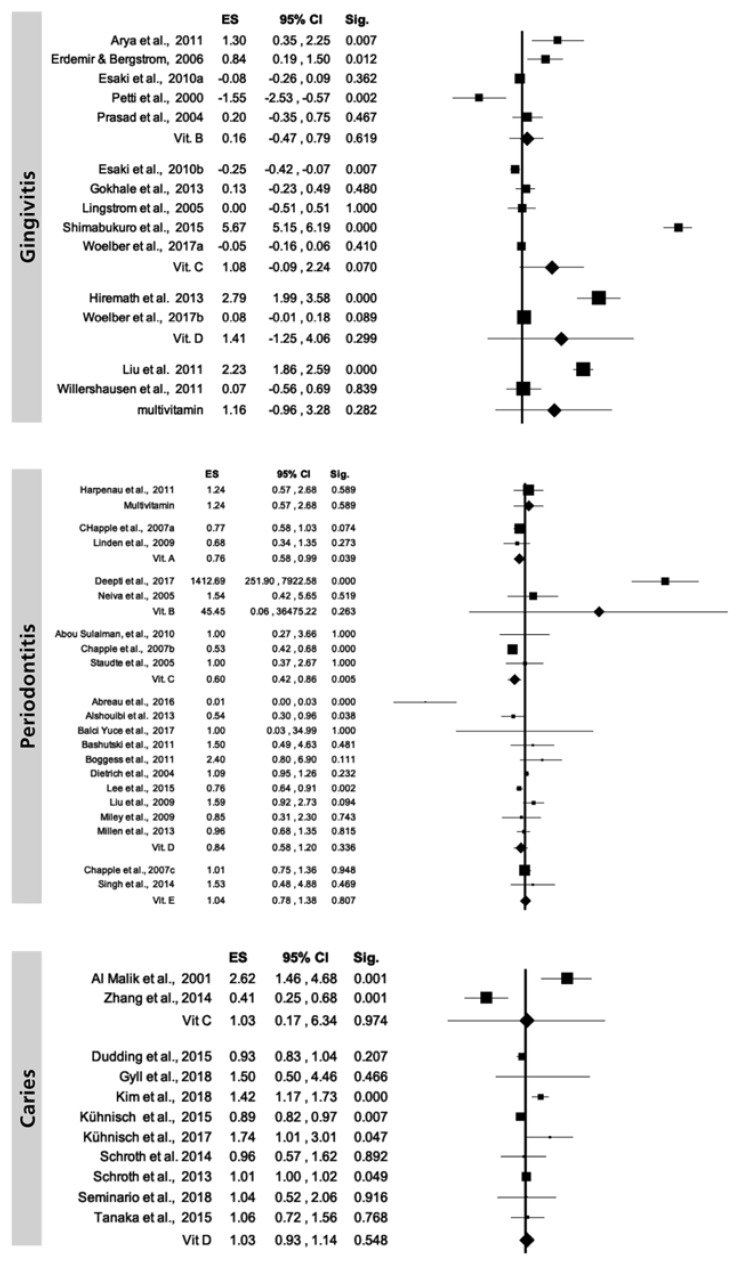
Random-effects model overall level of studies included, categorized by vitamins.

**Table 1 ijerph-17-00938-t001:** General characteristics of the studies included: (**a**) Gingivitis and periodontitis; (**b**) Hard dental tissues (dental caries, enamel defects).

Author	Sources	Type of Study	Vitamins	Oral Conditions	Quality Assessment
Li et al., [16]	Clin. Implant. Dent. Relat. Res. 2018, 20, 793–98	CT	Vit. C	Periodontitis	Good
Balci Yuce et al., [17]	J. Oral. Sci. 2017, 59,:397–404	CT	Vit. D	Periodontitis	Good
Deepti et al., [18]	J. Periodontol. 2017, 88, 999–1011	RCT	Vit. B7	Gingivitis/Periodontitis	Good
Abreu et al. [19]	BMC Oral Health 2016, 16, 89	CCS	Vit. D	Periodontitis	Fair
Adegboye et al., [20]	Public Health Nutr. 2016, 19, 503–51	CSS	Vit. D	Periodontitis	Good
Gümüş et al., [21]	Arch. Or. Biol. 2016, 63, 1–6	CS	Vit. D	Periodontitis	Good
Pavlesen et al., [22]	J. Periodontol 2016; 87, 852–63	RCS	Vit. D	Periodontitis/Tooth loss/Caries	Good
Woelber et al., [23]	BMC Oral Health 2016, 17, 28	RCT	Vit. C, D	Gingivitis/Periodontitis	Good
Shimabukuro et al., [24]	J. Periodontol 2015, 86, 27–35	RCT	Vit. C	Gingivitis	Good
Lee et al., [25]	Comm Dent. Oral Ep. 2015, 43, 471–8	CSS	Vit. D	Periodontitis	Good
Singh et al., [26]	J. Periodontol 2014, 85, 242–9	RCT	Vit. E	Periodontitis	Good
Jimenez et al., [27]	Public Health Nutr. 2014, 17, 844–52	CSS	Vit. D	Periodontitis/Tooth loss	Good
Alshouibi et al., [28]	J. Dent. Res. 2013, 92, 689–93	CSS	Vit. D	Periodontitis	Good
Gokhale et al., [29]	J. Diet. Suppl. 2013, 10, 93–104.	RCT	Vit. C	Periodontitis	Fair
Hiremath et al., [30]	Oral Health Prev. Dent. 2013, 11, 61–9	RCT	Vit. D	Gingivitis	Good
Iwasaki et al., [31]	Public Health Nutr 2013, 16, 330–38	RCS	Vit. A, D E, B6, B9, B12	Tooth loss	Good
Millen et al., [32]	J. Periodontol 2013, 84, 1243–56	CSS	Vit. D	Periodontitis	Good
Teles e t al., [33]	J. Periodontol 2012, 83, 1183–91	CSS	Vit. D	Periodontitis	Fair
Arya et al., [34]	Neurology 2011, 76,1338–43	RCT	Vit. B9	Gingivitis	Fair
Bashutski et al., [35]	J. Dent. Res. 2011, 90, 1007–12	RCT	Vit. D	Periodontitis	Good
Boggess et al., [36]	J. Periodontol 2011, 82, 195–200	CCS	Vit. D	Periodontitis	Good
Harpenau et al., [37]	J. Calif. Dent. Assoc. 2011, 39, 309–18	CT	Vit A, C, E, B6, B9, B12	Gingivitis	Poor
Liu et al., [38]	Asia Pac. J. Clin. Nutr., 2011, 20, 375–82	RCT	Vit. A, B1, B2, B6, B9, C, D, E	Gingivitis	Fair
Willershausen et al., [39]	Eur. J. Med. Res. 2011, 16, 514–18	CSS	Vit. B complex, C, D, E	Gingivitis	Fair
Abou Sulaiman, et al., [40]	J. Periodontol 2010, 81, 1547–54	RCT	Vit. C	Periodontitis	Fair
Esaki et al., [41]	Oral Disease 2010, 16,96–101	CSS	Vit. A, B1, B2 B9, C	Gingivitis	Good
Liu, et al., [42]	J. Periodontol 2009, 80, 1114–20	CT	Vit. D	Periodontitis	Fair
Llena et al., [43]	Quintessence Int. 2009, 40, 497–501	CT	Vit. B3, pro-vitamin B5	Gingivitis	Poor
Linden et al., [44]	J. Clin Periodontol. 2009, 36, 843–4	CSS	Vit. A	Periodontitis	Good
Miley et al., [45]	J. Periodontol. 2009, 80, 1433–39	CSS	Vit. D	Periodontitis	Good
Chapple, et al., [46]	J. Nutr. 2007, 137, 657–64	CSS	Vit. A, C, E	Periodontitis	Good
Dietrich et al., [47]	J. Dent. Res 2006, 85, 1134–37	CSS	Vit. C	Periodontitis	Good
Erdemir & Bergstrom [48]	J. Clin. Periodontol. 2006, 33, 878–84	CSS	Vit B9, B12	Periodontitis	Good
Lingstrom et al., [49]	Eur. J. Oral Sci. 2005, 113, 20–27	CT	Vit. C	Gingivitis	Good
Neiva et al., [50]	J. Periodontol 2005, 76, 1084–91	RCT	Vit. B complex	Periodontitis	Fair
Staudte et al., [51]	Br. Dent. J. 2005, 199, 213–7	CT	Vit. C	Gingivitis	Fair
Dietrich et al., [52]	Am. J. Clin. Nutr. 2004, 80, 108–13	CSS	Vit. D	Periodontitis	Good
Prasad et al., [53]	J. Indian Soc. Pedo Prev. Dent. 2004 22, 82–91	CT	Vit. B9	Gingivitis	Fair
Krall et al., [54]	Am. J. Med. 2001, 111, 452-456	RCT	Vit. D	Tooth loss	Good
Petti et al., [55]	Community Dent. Oral Epidemiol. 2000, 28, 407–413	CCS	Vit. A, B1, B2, B3, C	Gingivitis/Caries	Fair
(**a**)					
Syed et al., [56]	*BioMed. Res. Int.***2019**, 4503450	CSS	Vit. A, C, E	Caries	Poor
Gyll et al., [10]	*Nutr. J.***2018***, 17*, 11	CSS	Vit. D	Caries	Good
Kim et al., [57]	*BMC Oral Health***2018**, *18*, 43	CSS	Vit. D	Caries	Good
Seminario et al., [58]	*J. Dent. Child***2018**, *3*, 93–101	CSS	Vit. D	Caries	Good
van der Tas et al., [59]	*Community Dent. Oral Epidemiol.***2018**, *46*, 343-51	CS	Vit. D	Enamel defects	Good
Wójcik et al., [60]	*Medicine***2018**, *97*, 8(e9811)	CSS	Vit. D	Caries	Poor
Kühnisch et al., [61]	*Clin. Oral Invest.***2017**, *21*, 2283–2290	RCS	Vit. D	Caries	Good
Reed et al., [62]	*Ped. Dent. J.***2017**, *27*, 21e28	PiS	Vit. D	Enamel defects	Good
Pavlesen et al., [22]	*J Periodontol***2016**; 87, 852–63	CSS/PS	Vit. D	Periodontitis/Tooth loss/Caries	Good
Dudding et al., [63]	*PLoS One***2015**, 10, e0143769	MRS	Vit. D	Caries	Good
Kühnisch et al., [64]	*J. Dent. Res.***2015**, 94, 381–87	CSS	Vit. D	Enamel Defects	Good
Tanaka et al., [65]	*Annals of Epidemiology***2015**, *25*, 620e625	PSA	Vit. D	Caries	Good
Schroth et al., [66]	*Ped.***2014**, 133, e1277-e1284	RCS	Vit D	Caries	Fair
Zhang et al., [67]	*BMC Pub. Health***2014**, 14,7	CSS	Vit. C	Caries/Erosion	Fair
Schroth et al., [68]	*BMC Pediatrics***2013**, 13:174	CCS	Vit. D	Caries	Good
El Aidi et al., [69]	*Caries Res***2011**, 45, 303–312	PS	vitamins (unspecified)	Erosion	Good
MacKeown et al., [70]	*Community Dent Oral Epidemiol***2003**, *31*, 213–20	CSS	Vit. A, B complex, C, D	Caries	Good
Al Malik et al., [71]	*Int. J. Paed. Dent.***2001**, 11, 430–39	CSS	Vit. C	Caries/Erosion	Good
Aine et al., [72]	*J Oral Pathol Med***2000**, 29, 403–9	CCS	Vit. D	Enamel defects	Good
Petti et al., [55]	*Community Dent Oral Epidemiol***2000**; 28, 407–13	CCS	Vit. B12	Gingivitis/Caries	Fair
(**b**)					

**CCS**: Case-control study; **CS**: Cohort Study; **CSS**: Cross-Sectional Study; **CT**: Clinical Trial, **MRS**: Mendelian Randomization Study; **PSA**: Prospective Study Analysis; **PiS**: Pilot Study **RCS**: Retrospective Cohort Study; **RCT**: Randomized Clinical Trial.

**Table 2 ijerph-17-00938-t002:** Main characteristics of the studies included regarding gingivitis and periodontitis/tooth loss. (**a**) Gingivitis, (**b**) Periodontitis, (**c**) Tooth loss.

Author (Year)	Type of Study	Location	N Subjects Age-Range	M/F	Study Length	Vitamins Administration	Study Design (Groups Treatment)	Physical/Pathologic Condition	Outcomes
Gümüş et al., (2016) [21]	CS	USA	176 (19–40 yy)	F	---	Vit. D ---	3 groups: *-pregnancy (n = 59)* *-post-partum (n = 47)* *-non pregnant woman (n = 70)* Oral examination Vit. D in saliva	Pregnancy post-partum	In pregnancy and post-partum low level of Vit. D were statistically significantly associated to bleeding on probing
Shimabukuro et al., (2015) [24]	RCT	Japan	300 (20–64 yy)	M/F	3 months	Vit. C toothpaste	2 groups: *-toothpaste with Vit. C* *-control toothpaste*	None	Vit. C toothpaste statistically significantly reduced gingival inflammation (*p* < 0.01)
Woelber et al., (2017) [23]	RCT	Germany	15 (23–70 yy)	M/F	8 weeks	Vit. C, D ---	2 groups: -*diet modification* (n = 10) -*no diet modification* (n = 5)	None	In *diet modification* group gingival parameters improved (*p* < 0.05)
Gokhale et al., (2013) [29]	RCT	India	120 (30–60 yy)	M/F	2 weeks	Vit. C ---	4 groups: -*healthy subjects* *-chronic gingivitis* *-chronic periodontitis* *-chronic periodontitis and type 2 diabetes* Scaling and root planing with or without vit. C supplementation (450 mg)	Type 2 diabetes	A statistically significantly reduction in the bleeding score in the following groups that received Vit. C: *-chronic gingivitis* *-chronic periodontitis and type 2 diabetes*
Hiremath et al., (2013) [30]	RCT	India	110 (18–64 yy)	M/F	3 months	Vit. D tablets	4 groups: *-2000 UI Vit. D* *-1000 UI Vit. D* *-500 UI Vit. D* *-placebo*	None	Gingivitis scores improved: *-2000 UI Vit. D*: 2.4 at baseline to 0.3 at 3mo. *-1000 UI Vit. D*: 2.3 at baseline to 0.5 at 3 mo. *-500 UI Vit.* *D:* 2.2 at baseline to 0.8 at 3 mo. *-placebo:* 2.2 at baseline to 1.8 at 3 mo.
Arya et al., (2011) [34]	RCT	India	120 (6–15 yy)	M/F	6 months	Vit. B9 tablets	2 groups: *-Vit. B9 (5mg/die) (n = 62)* *-placebo (n = 68)* Oral examination	Epilepsy	*Vit. B9* group 21% developed PIGO *Placebo* group 88% developed PIGO
Liu et al., (2011) [38]	RCT	China	196 (54–72 yy)	M/F	6 months	Vit. A, B1, B2, B6, B9, C, D, E ---	2 groups: -*multivitamin formula (n = 97)* *-placebo (n = 99)*	Type 2 diabetes	Gingivitis and oral ulcer incidences were lower in *multivitamin formula* group (*p* < 0.05)
Willershausen et al., (2011) [39]	CSS	Germany	40 (24–30 yy)	M/F	3 months	Vit. B complex, C, D, E ---	2 groups: -*multivitamin formula (n = 19)* *-no treatment (n = 21)* Oral examination, dietary questionnaire, microbiology and blood analysis	None	A slight improvement of gingival inflammation in *micronutrients* group.
Esaki et al., (2010) [41]	CSS	Japan	497 (---)	M/F	---	Vit. A, B1, B2, B9, C ---	Oral examination Dietary questionnaire	None	Higher bleeding on probing scores statistically significantly associated to lower levels of Vit. C and Vit. B9 (*p* < 0.01) and Vit. A (*p* < 0.05)
Llena et al., (2009) [43]	CT	Spain	4 8(20–34 yy)	M/F	3 months +3 months	Vit. B3, Pro-vit. B5 toothpaste	2 groups (cross-over design): -*fluoridated toothpaste* -*fluoridated toothpaste with Vit. B3/Provit.* *B5,* Oral examination/calculus presence	None	Fluoridated toothpaste with Vit. B3/Provit. B5 provided a statistically significant reduction in calculus presence (*p* = 0.01)
Erdemir &Bergstrom, (2006) [48]	CSS	Sweden	88(30–69 yy)	M/F	---	Vit. B9, B12 ---	2 groups: *-current smokers (n = 45)* *-non smokers (n = 43)* Oral examination/Vit. B9, B12 serum level	None	In smokers higher gingival index scores and lower Vit. B9 levels (*p* < 0.05 for both)
Lingstrom et al., (2005) [49]	CT	Sweden	30 (---)	M/F	3 months	Vit. C gums	2 groups: *-5 pieces/day chewing gum vit. C/without vit. C, no gum use* *-10 pieces/day chewing gum with vit.* *C+ carbamide (30 mg + 30 mg), no gum use* Calculus score/plaque/gingivitis	None	A significant reduction in the total calculus score after the use of Vit. C (33%) and Vit. C + carbamide (12%) gums compared with no gum use
Prasad et al., (2004) [53]	CT	India	60 (8–13 yy)	M/F	1 year	Vit. B9 tablets	2 groups: *-Vit. B9 (5mg/die) and oral hygiene instruction* *-oral hygiene instructions alone* Oral examination	Epilepsy treated with phenytoin	Gingival overgrowth: 60% in *oral hygiene instruction alone* and 50% in *Vit. B9 and oral hygiene instruction*. Delay in onset of overgrowth in *Vit. B9 and oral hygiene instruction*
Petti et al., (2000) [55]	CCS	Italy	54 (17–19 yy)	F	---	Vit. A, B1, B2, B3, C ---	2 groups: -*gingivitis affected* *-no gingivitis affected* Oral examination/ three-day food record	None	Vit. B1 and Vit. B2 levels statistically lower in subjects with gingivitis presence
(**a**)									
Balci Yuce et al., (2017) [17]	CT	Turkey	53 (37–61 yy)	M/F	6 weeks	Vit. D ---	3 groups: -*rheumatoid arthritis/periodontitis (RP)* *-periodontitis (P)* *-healthy (H)* Treatment: initial periodontal treatment	Rheumatoid arthritis	Periodontal parameters statistically significantly improved in all groups Vit. D was higher in *RP* and *P* than in *H* group and decreased in RP group after treatment
Deepti et al., (2017) [18]	RCT	India	60 (15–34 yy)	F	3-6 months	Vit. B7 ---	2 groups: *- scaling-root planing+Vit. B7* *- Vit. B7*	Polycystic ovary syndrome (PCOS)	In *scaling-root planing+Vit. B7* group a statistically significantly reduction of C-reactive protein and insulin resistance at 3-6 mo. was found. Periodontal parameters also improved at 3-6 mo.
Abreu et al. (2016) [19]	CCS	Puerto Rico	4 8(35-64 yy)	M/F	---	Vit. D ---	2 groups: -*moderate/severe periodontitis* *-healthy*	None	Lower OR for periodontitis (OR = 0.885; 95%CI= 0.785, 0.997) for each Vit. D unit increase
Adegboye et al., (2015) [20]	CSS	Denmark	3287 (18–95 yy)	M/F	---	Vit. D ---	Dietary questionnaire Oral examination	None	No association Vit. D levels between severe periodontitis presence
Lee et al., (2015) [25]	CSS	Korea	6011 (---)	M/F	---	Vit. D ---	Vit. D level Oral examination (CPI index)	None	No association between Vit. D level and periodontitis In smokers an association was found (OR 1.53, 95% CI 1.07–2.18)
Singh et al., (2014 [26])	RCT	India	60 (22–50 yy)	M/F	3 months	Vit. E tablets	2 groups: -*periodontitis (n = 38): 19 treated with scaling/root planing (SRP) and 19 with SRP+300 IU Vit. E* *-healthy (n = 22) no treatment*	None	Superoxide dismutase improved in both treatment groups, but was higher in *SRP+300 IU Vit. E* (*p* < 0.05)
Alshouibi et al. (2013) [28]	CS	USA	562 (---)	M	---	Vit. D ---	Vit. D intake Oral examination (4 times during 12 yy)	None	Vit. D intake ≥ 800 IU associated with lower odds of severe periodontal disease (OR = 0.67, 95% CI = 0.55-0.81)
Gokhale et al., (2013) [29]	RCT	India	120 (30–60 yy)	NA	2 weeks	Vit. C ---	4 groups -*healthy subjects;* *-chronic gingivitis;* *-chronic periodontitis,* *-chronic periodontitis and type 2 diabetes* Scaling and root planing with or without vit. C supplementation (450 mg)	Type 2 diabetes	Statistically significant reduction of the bleeding score in the subgroups receiving Vit. C
Millen et al., (2013) [32]	CSS	USA	920 (50–79 yy)	F	---	Vit. D ---	*-Vit. D level* *-oral examination*	Post-menopausa	No association Vit. D and alveolar crestal height/ tooth loss OR = 0.96, (95%CI: 0.68–1.35). Vit. D associated to clinical attachment level and probing pocket depth (95%CI: 5–53%)
Teles et al., (2012 [33])	CSS	USA	56 (23–71 yy)	M/F	6 months	Vit. D ---	Periodontal patients Scaling, root planing and hygiene instruction Bacteria in sub-gingival plaque Vit. D level	None	No associations between Vit. D and clinical and microbial parameters
Bashutski et al., (2011) [35]	RCT	USA	40 (31–65 yy)	M/F	6 months	Vit. D tablets	2 groups: *-periodontal surgery, Ca (1000 mg) and Vit. D (800 UI) supplements and self-administered teriparatide for 6 weeks* - *periodontal surgery, Ca (1000 mg) and Vit. D (800 UI) supplements and placebo for 6 weeks*	None	Vit. D levels had no statistically significant impact on clinical attachment level and probing pocket depth improvements in teriparatide patients
Boggess et al., (2011) [36]	CCS	USA	233 (21–33 yy)	F	---	Vit. D ---	2 groups: -*pregnant woman with moderate to severe periodontitis* *-pregnant woman without periodontitis* Vit. D level and Oral examination between 14 and 26 weeks of gestation	Pregnancy	Pregnant woman with periodontitis had statistically significant lower Vit. D levels and more likely to have Vit. D insufficiency (65% *versus* 29%)
Harpenau et al., (2011) [37]	CT	USA	89 (18–70 yy)	M/F	8 weeks	Vit. A, C, E, B6, B9, B12 tablets	2 groups with mild to severe periodontitis: *-multivitamin formula* *-placebo*	None	Both groups showed non-significant trends for improvement in gingival, bleeding, probing depth and clinical attachment scores.
Abou Sulaiman, et al., (2010) [40]	RCT	Syria	60 (23–65 yy)	M/F	3 months	Vit. C tablets	2 groups: *-chronic periodontitis (n = 30)-15 subjects non-surgical treatment plus Vit. C and 15 subjects non surgical treatment alone* *-healthy controls(n = 30)*	None	The two groups showed significant reductions in all clinical measures
Liu, et al., (2009) [41]	CT	China	17 8(23–41 yy)	M/F	---	Vit. D ---	3 groups: *-aggressive periodontitis (AgP) (n = 66)* *-chronic periodontitis (CP) (n = 52)* *-healthy controls (n = 60)* Oral examination/Vit. D level	None	In *AgP* Vit. D was higher than in *healthy controls* (29.28 vs. 21.60 nmol/l; *p* < 0.05) and significantly correlated with bleeding index (*r* = 0.321; *p* < 0.05).
Linden et al., (2009) [44]	CSS	United Kingdom	125 8(60–70 yy)	M	---	Vit. A---	Oral examination/questionnaire	None	Vit. A lower in the men with low-threshold periodontitis (*p* < 0.001) and high-threshold periodontitis (*p* = 0.002) compared to subjects without periodontitis
Miley et al., (2009) [45]	CSS	USA	51 (50–80 yy)	M/F	---	Vit. D tablets	2 groups: *-periodontal maintenance + Vit. D (400 IU/day) and Ca (1,000 mg/day) (n = 23)* *-periodontal maintenance only (n = 28)* Oral examination	None	*Periodontal maintenance + Vit. D* (400 IU/day) *and Ca* (1000 mg/day) had lower but not statistically significant probing depths
Chapple, et al., (2007) [46]	CSS	USA	11,480 (25–70 yy)	M/F	---	Vit. A, C, E ---	Oral examination/questionnaire/ Vit. A, C, E level	None	Subjects with the highest values of serum Vit. C had 47% (95%CI 32, 58) lower odds of periodontitis than subjects with the lowest values (trend OR: 0.76, 95%CI 0.69, 0.84)
Dietrich et al., (2006) [47]	CSS	USA	462 (47–92 yy)	M	---	Vit. C ---	2 groups: *-periodontitis (n = 86)* *-no periodontitis (n = 376)* Oral examination/Dietary questionnaire	None	Subjects with periodontitis had a Vit. C intake (mg) lower than subjects without periodontitis (120±201 *vs* 197±267)
Neiva et al., (2005) [50]	RCT	USA	30 (38–65 yy)	M/F	180 days	Vit. B complex tablets	2 groups: *- periodontal surgery and Vit-B for 30 days* *- periodontal surgery and placebo* Oral examination/BANA test	None	Statistically significant difference between mean clinical attachment level between *periodontal surgery and Vit. B* (+0.41±0.12) and *periodontal surgery and placebo* (−0.52 ± 0.23)
Staudte et al., (2005) [51]	CT	Germany	80 (22–75 yy)	M/F	2 weeks	Vit. C diet	2 groups: *-healthy (n = 22)* *-periodontitis: using or not grapefruit (n = 38)* Oral examination/Vit. C level	None	Bleeding index statistically significantly decreased after Grapefruit consumption
Dietrich et al., (2004) [52]	CSS	USA	11,202 (20–75 yy)	M/F	---	Vit. D ---	Oral examination/Vit. D level	None	Vit. D levels were statistically significantly lower in men/women >50 years with greater periodontal attachment loss
(**b**)									
Pavlesen et al., (2016) [22]	CSS/PS	USA	70 (53–85 yy)	F	5 years	Vit. D ---	Oral examination Vit. D level	Post-menopausa	No association between Vit. D levels and history or incidence of tooth loss caused by periodontal disease
Jimenez et al., (2014) [27]	CSS	USA	42,730 (40–75 yy)	M	---	Vit. D ---	Self-reported tooth loss and periodontitis Predicted Vit. D based on data on 1095 man	None	Men with highest levels of Vit. D exhibited a significantly lower risk of tooth loss compared with men with lowest levels
Iwasaki et al., (2013) [31]	RCS	Japan	286 (75–80 yy)	M/F	5 years	Vit. A, D E, B6, B9, B12 tablets	Dietary questionnaire (baseline and after 5 yy) Oral examination (functional tooth units)	None	Subjects with impaired dentition showed a significantly greater decline in nutrients intake (Vit. A and E)
Millen et al., (2013) [32]	CSS	USA	920 (50–79 yy)	F	---	Vit. D ---	Vit. D level Oral examination	Post-menopausa	No association between Vit. D and alveolar crestal height/ tooth loss *OR* = 0.96, (95%CI: 0.68–1.35)
(**c**)									

**95%CI**: Confidence Intervals; **CCS**: Case-control study; **CS**: Cohort Study; **CSS**: Cross-Sectional Study; **CT**: Clinical Trial, **MRS**: Mendelian Randomization Study; **PIGO**: Phenytoin-induced gingival overgrowth; **PS**: Pilot study; **PSA**: Prospective Study Analysis; **RCS**: Retrospective Cohort Study; **RCT**: Randomized Clinical Trial; **F**: Females; **M**: Males; **OR**: Odds Ratio; **yy**: age range in years.

**Table 3 ijerph-17-00938-t003:** Main characteristics of the studies included regarding hard dental pathological processes ((**a**) caries and (**b**) enamel defects).

Author (Year)	Type of Study	Location	N Subjects Age-Range	M/F	Study Length	Vitamins Administration	Study Design (Groups Treatment)	Physical/Pathologic Condition	Outcomes
Syed et al., (2019) [56]	CSS	Saudi Arabia	100 (6–13 yy)	M/F	---	Vit. A, C, E ---	Two groups: *-DMFS/dmfs=0* *-DMFS/dmfs>3* Saliva samples	None	Vit. A not statistically significantly correlated Vit. C and Vit. E. statistically significantly correlated to caries experience
Gyll et al., (2018) [10]	CSS	Sweden	206 (8 yy)	M/F	---	Vit. D ---	Vit. D serum level DMFT	None	Vit. D level statistically significantly associated to caries experience (*OR* = 0.96; *p* = 0.024)
Kim et al., (2018) [57]	CSS	Korea	1688 (10-11 yy)	M/F	---	Vit. D ---	Vit. D serum level DMFT	None	*Vit. D* < 0.25 nmol statistically significantly associated to caries experience (*p* < 0.05)
Seminario et al., (2018) [58]	CSS	USA	276 (1–6 yy)	M/F	---	Vit. D ---	Vit. D serum level DMFT	Neurologic and genetic disabilities	Vit. D level associated to caries in neurologic (*p* < 0.01) and genetic (*p* < 0.01) conditions
Wójcik et al., (2018) [60]	CS	Poland	121 (6–17 yy)	M/F	9 months	Vit. D ---	Human recombinant growth hormone Vit. D serum level DMFT	Growth problems	Caries prevalence reduced by 0.66 per each 10 ng/mL of Vit. D increase
Kühnisch et al., (2017) [61]	RCS	Germany	406 (at birth)	M/F	10 yy	Vit. D tablets	Vit. D supplementation Fluoride varnish dmfs	None	Vit. D + fluoride < 6 months associated to caries *OR* = 2.47 (95%CI = 1.32–4.63) Vit. D + fluoride >6 mo. associated to caries *OR* = 2.08 (95%CI = 1.00–4.32)
Pavlesen et al., (2016) [22]	CSS/PS	USA	558 (53–85 yy)	F	5 yy	Vit. D ---	Oral examination Vit. D serum level	Post-menopausa	Tooth loss due to caries associated to Vit. D *OR* = 1.03 (95%CI =0.62/1.72)
Dudding et al., (2015) [63]	MRS	UK	5545 (3–8 yy)	M/F	---	Vit- D ---	Vit. D serum level DMFT/dmft	None	Caries not statistically significantly associated to 10 ng/mL of Vit. D increase OR 0.93 (95%CI = 0.83-1.05)
Kühnisch et al., (2015) [64]	CSS	Germany	1148 (--)	M/F	10 yy	Vit. D tablets	Vit. D supplementation DMFT/dmft MIH	None	Vit. D statistically significantly associated to caries OR 0.90 (95%CI = 0.82–0.97) per each 10 ng/mL of Vit. D increase
Tanaka et al., (2015) [65]	PSA	Japan	1210 mother-child (36–46 mo)	M/F	---	Vit. D diet	Vit. D intake in pregnancy dmft in children	None	OR for dmft 1.06 (95%CI = 0.72–1.56) of Vit. D during pregnancy, quartiles 2, 3 OR for dmft 0.67 (95%CI = 0.44–1.22) of Vit. D during pregnancy, quartiles 4.
Schroth et al., (2014) [66]	RCS	Canada	207 mother-child (--)	M/F	---	Vit. D---	Serum Vit. D in pregnancy ECC Enamel defects	None	Low serum Vit. D in pregnancy was statistically significantly associated ECC experience
Zhang et al., (2014) [67]	CSS	Hong Kong	600 (12 yy)	M/F	1 yy	Vit. C tablets	Vit, C intake DMFT BEWE	None	OR for caries experience 1.75 (95%CI = 1.14–2.69) in girls Vit. C supplements
Schroth et al., (2013) [68]	CCS	Canada	266 (--)	M/F	2 yy	Vit. D ---	Serum level Vit. D ECC	None	Low Vit. D associated to high ECC levels
MacKeown et al., (2003) [70]	CSS	South Africa	259 (2–5 yy)	M/F	4 yy	Vit. A, B complex, C, D diet	Vitamins intake dmft	None	Vit. B2, B7, B12 statistically significantly associated to caries incidence
Al Malik et al., (2001) [71]	CSS	Saudi Arabia	987 (2–5 yy)	M/F		Vit. C ---	Vit. C intake dmft/s, Tooth Erosion	None	Vit. C intake not statistically significantly significant associated to caries.
(**a**)									
van der Tas et al., (2018) [59]	CS	the Netherlands	4750 (6 yy)	M/F	---	Vit. D ---	Foetal Vit. D MIH at 6 yy	None	Vit. D not statistically significantly associated to MIH
Reed et al., (2017) [62]	PS	USA	37 (--)	M/F	---	Vit. D tablets	Vit. D supplementation (first year of age)	None	Maternal pregnant Vit. D level statistically significantly associated to Enamel hypoplasia in children
Kühnisch et al., (2017) [61]	RCS	Germany	406 (at birth)	M/F	10 yy	Vit. D tablets	Vit. D supplementation Fluoride varnish MIH	None	MIH not statistically significantly associated with Vit. D + Fluoride <6 mo. OR = 1.71 (95%CI = 0.67–4.38) and Vit. D + fluoride >6 mo. OR = 0.57 (95%CI = 0.21–1.55)
Kühnisch et al., (2015) [64]	CSS	Germany	1148 (--)	M/F	10 yy	Vit. D tablets	Vit. D supplémentation dmft/DMFT MIH	None	MIH statistically significantly associated to 10 ng/mL of Vit. D increase OR 0.89 (95%CI = 0.82–0.97)
Schroth et al., (2014) [66]	RCS	Canada	207 mother-child (--)	M/F	---	Vit. D ---	Serum Vit. D in pregnancy ECC Enamel defects	None	Low serum Vit. D in pregnancy was statistically significantly associated ECC experience
Zhang et al., (2014) [67]	CSS	Hong Kong	600 (12 yy)	M/F	1 yy	Vit. C diet	Vit. C intake DMFT BEWE	None	Tooth erosion not statistically significantly associated to frequency of Vit. C supplement drinks (*p* = 0.064)
El Aidi et al., (2011) [69]	PS	the Netherlands	572 (10–12 yy)	M/F	---	Vitamins (unspecified) tablets	Vitamins intake Tooth Erosion	None	Vitamins intake statistically significantly associated to erosion progression OR = 2.03 (95%CI = 1.14–3.62)
Al Malik et al., (2001) [71]	CSS	Saudi Arabia	987 (2–5-yy)	M/F	---	Vit. C ---	Vit. C intake dmft/s, Tooth Erosion	None	Vit. C intake statistically significantly associated to erosion.
Aine et al., (2000) [72]	CCS	Finland	96 (1–2 yy)	M/F	---	Vit. D tablets	Vit. D supplementation Enamel defects	Preterm children	Vit. D supplementation was not statistically significantly enamel defects.
(**b**)									

**DMFT/DMFS**: Decayed, Missing, Filled Tooth/Surfaces index in permanent teeth; **dmft/dmfs**: decayed, missing, filled footh/surfaces index in primary teeth; **MIH**: Molar Incisor Hypomineralization; **ECC**: Early Childhood Caries; **BEWE**: Basic Erosive Wear Examination; **95%CI**: Confidence Interval; **OR**: Odds Ratio; **yy**: age range in years.

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
