# Peer review of "The Role of Vitamins in Oral Health. A Systematic Review and Meta-Analysis"

_ijerph, 2020, doi:10.3390/ijerph17030938_

Round 1

Reviewer 1 Report

The authors have provided a detailed review of the concerned topic. However, there are some concerns that need to be addressed.

Introduction:

Line 33: Change the word ‘clarify’ to ‘clarified’.

Line 36: Add such after vitamins in the sentence starting from ‘fat soluble vitamins……’

Materials and Methods:

In the eligibility criteria, please specify if only electronically published articles were taken into consideration or paper-based articles were also looked at.

I would prefer if the authors italicize the search terms (MeSH terms and key words).

The data collection, summary measures and synthesis of results section is not clearly written. Please specify that the first paragraph is discussing the outcome measure and that the outcome measure is measured by having the following characteristics……… then proceed with defining the measures.

Similar is the case with the exposure variable – use of vitamins. Discuss the form of administration of vitamins – liquids, tablets etc… how was it measured?

In the assessment of bias, please discuss in brief the quality assessment tool and how the papers were ranked based on that tool.

Results:

Line 129: Omit the word ‘finally’.

In Table 2, under the outcome column, please clarify if there is positive or negative association, for example, in the study by Gumus et al, it is written that that vitamin D is statistically significantly associated to bleeding on probing. It is not clear by this sentence whether the use or deficiency of Vitamin D is associated with bleeding to probing. Please clarify this. Similar is the case for a few other papers in the table, Esaki et al, Petti et al, Lee et al and many more. Please be clear on all of these. Table 3 also have similar issues. Please correct them.

There are a ton of grammatical and spelling mistakes in table 2 under ‘outcomes’ column. Please make sure that the sentences are framed in a well-structured form and there are no spelling mistakes.

Line 192: Add ‘who were’ after similar characteristics and remove the comma.

Be consistent with the way, Vitamin is written. In some sentences it is written as Vit. A or C or…… whereas at other places, it is written with a small ‘v’ as vit. A or sometimes at Vitamin C or A. Please choose one format and make sure that it is consistent throughout the paper including tables as well.

Line 200 and 206: The phrase ‘resulted not associated’ is not a correct grammatical form. Please rewrite the sentence. It could be written as, ‘while Vit……. was not associated with ………..’

Line 211: Incorrect word ‘statically’. Change it to statistically. Please be careful with the spellings, there are a few spelling mistakes in this paper.

Line 216, 219, table 3, 239, 240: It should be ‘statistically significantly lower’ opposed to ‘statistically significant lower’. Please make sure that ‘statistically significant lower or superior or correlated’ is changed to ‘statistically significantly lower or superior or correlated’ throughout the manuscript unless the sentence is structured differently.

Line 222: Explain in detail the about the studies. What was reported, what periods of women’s life were looked at.

Line 226: Capitalize the first alphabet H.

Line 248: Rewrite the sentence beginning from ‘salivary vitamin A…….’

The manuscript is missing a major section, “DISCUSSION”. Discussion is the most important section in a manuscript. The authors have not discussed their findings, what could be the reason behind the results of the study. If they have found a negative association, what could be the literature behind it? What do other similar studies say about the findings? What could be the impact of these findings? All these points need to be discussed. The authors need to do an extensive literature review to discuss the findings of their systematic review. The authors should read other systematic reviews to get into insights of discussion section of systematic reviews.

Thank you.

Author Response

We vuole like to thank the reviewer for the comments and suggestions the helped us to improve our manuscript.

Here a point-by point response to the reviewer's comments. 0ur replies are in italic.

Introduction:

Line 33: Change the word ‘clarify’ to ‘clarified’.

Done

Line 36: Add such after vitamins in the sentence starting from ‘fat soluble vitamins……’

Done

Materials and Methods:

In the eligibility criteria, please specify if only electronically published articles were taken into consideration or paper-based articles were also looked at.

Done

I would prefer if the authors italicize the search terms (MeSH terms and key words).

Done

The data collection, summary measures and synthesis of results section is not clearly written. Please specify that the first paragraph is discussing the outcome measure and that the outcome measure is measured by having the following characteristics……… then proceed with defining the measures.

We modified the data collection paragraph following the reviewer’s suggestion.

Similar is the case with the exposure variable – use of vitamins. Discuss the form of administration of vitamins – liquids, tablets etc… how was it measured?

Following the reviewer’s comment, we add the administration of vitamins to both tables 2 and 3.

In the assessment of bias, please discuss in brief the quality assessment tool and how the papers were ranked based on that tool.

We briefly discuss, the assessment of bias and the quality assessment toll following reviewer’s suggestion.

Results:

Line 129: Omit the word ‘finally’.

Done

In Table 2, under the outcome column, please clarify if there is positive or negative association, for example, in the study by Gumus et al, it is written that that vitamin D is statistically significantly associated to bleeding on probing. It is not clear by this sentence whether the use or deficiency of Vitamin D is associated with bleeding to probing. Please clarify this. Similar is the case for a few other papers in the table, Esaki et al, Petti et al, Lee et al and many more. Please be clear on all of these. Table 3 also have similar issues. Please correct them.

We modified both table 2 and 3 following the reviewer’s comment.

There are a ton of grammatical and spelling mistakes in table 2 under ‘outcomes’ column. Please make sure that the sentences are framed in a well-structured form and there are no spelling mistakes.

We checked table 2.

Line 192: Add ‘who were’ after similar characteristics and remove the comma.

We modified the phrase as follows “The administration of a dietary supplement containing different micronutrients (including Vit. D, C, E, B complex) for 3 months produced a slight improvement of the gingival inflammation in students under stress with poor oral hygiene, compared to students also under stress but not provided with the dietary supplement.

Be consistent with the way, Vitamin is written. In some sentences it is written as Vit. A or C or…… whereas at other places, it is written with a small ‘v’ as vit. A or sometimes at Vitamin C or A. Please choose one format and make sure that it is consistent throughout the paper including tables as well.

We tried to be consistent and we had decided to use the term Vitamin or Vit. when a specific vitamin is indicated.

Line 200 and 206: The phrase ‘resulted not associated’ is not a correct grammatical form. Please rewrite the sentence. It could be written as, ‘while Vit……. was not associated with ………..’

Done

Line 211: Incorrect word ‘statically’. Change it to statistically. Please be careful with the spellings, there are a few spelling mistakes in this paper.

Done

Line 216, 219, table 3, 239, 240: It should be ‘statistically significantly lower’ opposed to ‘statistically significant lower’. Please make sure that ‘statistically significant lower or superior or correlated’ is changed to ‘statistically significantly lower or superior or correlated’ throughout the manuscript unless the sentence is structured differently.

Done

Line 222: Explain in detail the about the studies. What was reported, what periods of women’s life were looked at.

Done

Line 226: Capitalize the first alphabet H.

Done

Line 248: Rewrite the sentence beginning from ‘salivary vitamin A…….’

The phrase was modified as follows “Salivary Vitamin A levels are not to be statically significantly associated to caries risk”.

The manuscript is missing a major section, “DISCUSSION”. Discussion is the most important section in a manuscript. The authors have not discussed their findings, what could be the reason behind the results of the study. If they have found a negative association, what could be the literature behind it? What do other similar studies say about the findings? What could be the impact of these findings? All these points need to be discussed. The authors need to do an extensive literature review to discuss the findings of their systematic review. The authors should read other systematic reviews to get into insights of discussion section of systematic reviews.

We followed the reviewer’s suggestion and we had implemented the discussion part. Usually systematic reviews and metanalysis papers did not include a proper discussion part  just a conclusion (i.e. Hujoel, P.P. Vitamin D and dental caries in controlled clinical trials: systematic review and meta-analysis. Nutrition Reviews 2012, 71, 88–97.)

Reviewer 2 Report

The abstract: well defined and structured and clearly defines the objectives, method results and conclusion of the study. 

Introduction: Well organised. But i think author needs to elaborate on the prevalence of vitamin deficiency related oral diseases worldwide, with a particular focus on gum disease (periodontitis) and dental erosion.

Methodology:

Well structured, Followed the PRISMA guidelines. Was the review registered with the PROSPERO?

Results:

The author describes the "Regarding gingival/periodontal disease, twenty-six papers 136 were ranked of as being of good quality, twelve were classified of fair quality and only two of poor quality. 137 Regarding hard dental tissues, sixteen papers were ranked of as being of good quality, four were classified of 138 fair quality and only two of poor quality". The references need to be added next to each statement claimed

There is no defined heading of discussion. Please add.

Conclusion: get rid of the first paragraph of this section. You have already said that in the introduction in your aims.

Author Response

We followed all the reviewer's comments-

Here point-by-point the response to reviewer's comments. Our replies are in italic.

Introduction: Well organised. But i think author needs to elaborate on the prevalence of vitamin deficiency related oral diseases worldwide, with a particular focus on gum disease (periodontitis) and dental erosion.

We really thank the reviewer for the comment. In literature, the correct scientific data on the prevalence of vitamin deficiency related oral diseases are almost absent. This was one the reason behind our decision to design and carried out this project. There is a “general knowledge” on the effect of vitamin deficiencies on oral health but without a scientific evidenceAnyway, we tried to modify the introduction section.

Methodology:

Well structured, Followed the PRISMA guidelines. Was the review registered with the PROSPERO?

Yes, the review was registered with the PROSPERO system but at the time of the first submission we did not received yet the registration number. (ID 150613  Sept 12th 2019)

Results:

The author describes the "Regarding gingival/periodontal disease, twenty-six papers 136 were ranked of as being of good quality, twelve were classified of fair quality and only two of poor quality. 137 Regarding hard dental tissues, sixteen papers were ranked of as being of good quality, four were classified of 138 fair quality and only two of poor quality". The references need to be added next to each statement claimed

Done

There is no defined heading of discussion. Please add.

Conclusion: get rid of the first paragraph of this section. You have already said that in the introduction in your aims.

Done. We followed the reviewer’s suggestion and we had implemented the discussion part. Usually systematic reviews and metanalysis papers did not include a proper discussion part  just a conclusion (i.e. Hujoel, P.P. Vitamin D and dental caries in controlled clinical trials: systematic review and meta-analysis. Nutrition Reviews 2012, 71, 88–97.)

Reviewer 3 Report

This is a meticulously done systematic review with good statistical analyses.

However, there are several instances where the syntax/grammar needs to be worked on. May be some professional editing will help.

There has been previously done a systematic review by Pachava et al "The Role of Vitamins and Trace Elements on Oral Health: A Systematic Review" in 2017 and the authors have not discussed the uniqueness of their review and how different is theirs from the 2017 article. (reviewer has no affiliation to the published article)

The discussion/conclusion is really short and needs elaboration. It lacks authors' views on the topic, the significance of their review, and their contribution to the field. It is not been conveyed. 

Author Response

We thanks the reviews for the positive comments.

Here is the point-by-point response to reviewer's comments. Our replies are in italic.

However, there are several instances where the syntax/grammar needs to be worked on. May be some professional editing will help.

We send the paper for a professional editing and language check.

There has been previously done a systematic review by Pachava et al "The Role of Vitamins and Trace Elements on Oral Health: A Systematic Review" in 2017 and the authors have not discussed the uniqueness of their review and how different is theirs from the 2017 article. (reviewer has no affiliation to the published article).

The review was added to reference list, even it very different from our: a- is not a systematic review nut a narrative one; - it is based to quite few papers; -gingival/periodontal disease is not included in the revision.

The discussion/conclusion is really short and needs elaboration. It lacks authors' views on the topic, the significance of their review, and their contribution to the field. It is not been conveyed. 

We followed the reviewer’s suggestion and we had implemented the discussion part. Usually systematic reviews and metanalysis papers did not include a proper discussion part  just a conclusion (i.e. Hujoel, P.P. Vitamin D and dental caries in controlled clinical trials: systematic review and meta-analysis. Nutrition Reviews 2012, 71, 88–97.)

Round 2

Reviewer 1 Report

The revised manuscript reads fine. The authors have made substantial changes based on my comments.

Author Response

We thank the reviewer for the aid to improve our paper.

The manuscript was again revised by an English native speaker

Reviewer 3 Report

Syntax/grammar still needs to be addressed 

( esp in the introduction and more obvious in the acknowledgment sections...)

Author Response

We really thank the reviewer for the comments.

We checked again the introduction and and the acknowledgement section.